# The Desmoid Dilemma: Challenges and Opportunities in Assessing Tumor Burden and Therapeutic Response

**DOI:** 10.3390/curroncol32050288

**Published:** 2025-05-21

**Authors:** Yu-Cherng Chang, Bryan Nixon, Felipe Souza, Fabiano Nassar Cardoso, Etan Dayan, Erik J. Geiger, Andrew Rosenberg, Gina D’Amato, Ty Subhawong

**Affiliations:** 1Department of Radiology, Sylvester Comprehensive Cancer Center, University of Miami Miller School of Medicine, Miami, FL 33136, USA; ychang40@umiami.edu (Y.-C.C.); bfn6@med.miami.edu (B.N.); ffs8@med.miami.edu (F.S.); fxn116@med.miami.edu (F.N.C.); exd864@miami.edu (E.D.); 2Department of Orthopaedics, Sylvester Comprehensive Cancer Center, University of Miami Miller School of Medicine, Miami, FL 33136, USA; egeiger@miami.edu; 3Department of Pathology, Sylvester Comprehensive Cancer Center, University of Miami Miller School of Medicine, Miami, FL 33136, USA; arosenberg@med.miami.edu; 4Department of Medicine, Sylvester Comprehensive Cancer Center, University of Miami Miller School of Medicine, Miami, FL 33136, USA; gina.damato@med.miami.edu

**Keywords:** desmoid tumor, MRI, radiomics, biomarkers, RECIST

## Abstract

Desmoid tumors are rare, locally invasive soft-tissue tumors with unpredictable clinical behavior. Imaging plays a crucial role in their diagnosis, measurement of disease burden, and assessment of treatment response. However, desmoid tumors’ unique imaging features present challenges to conventional imaging metrics. The heterogeneous nature of these tumors, with a variable composition (fibrous, myxoid, or cellular), complicates accurate delineation of tumor boundaries and volumetric assessment. Furthermore, desmoid tumors can demonstrate prolonged stability or spontaneous regression, and biologic quiescence is often manifested by collagenization rather than bulk size reduction, making traditional size-based response criteria, such as Response Evaluation Criteria in Solid Tumors (RECIST), suboptimal. To overcome these limitations, advanced imaging techniques offer promising opportunities. Functional and parametric imaging methods, such as diffusion-weighted MRI, dynamic contrast-enhanced MRI, and T2 relaxometry, can provide insights into tumor cellularity and maturation. Radiomics and artificial intelligence approaches may enhance quantitative analysis by extracting and correlating complex imaging features with biological behavior. Moreover, imaging biomarkers could facilitate earlier detection of treatment efficacy or resistance, enabling tailored therapy. By integrating advanced imaging into clinical practice, it may be possible to refine the evaluation of disease burden and treatment response, ultimately improving the management and outcomes of patients with desmoid tumors.

## 1. Introduction

Desmoid tumors (DTs), also known as aggressive fibromatosis, are rare, locally invasive soft-tissue clonal neoplasms. Although DTs do not metastasize, they can demonstrate a wide spectrum of clinical outcomes, including spontaneous regression, indolence, or potentially life-threatening organ involvement. Even when not lethal, DTs can be a source of debilitating pain and deformity.

At present, treatment is largely based on symptomatic presentation. Observation is suggested for asymptomatic patients. Symptomatic patients are commonly referred for medical therapy, and less frequently for surgery or radiotherapy given the predilection for postsurgical and postradiation recurrence and morbidity [1]. Medical therapy can be further divided among NSAIDs, hormonal anti-estrogen therapy, chemotherapy, and more recently, targeted therapy. In general, targeted therapies, such as tyrosine kinase inhibitors and the recently FDA-approved γ-secretase inhibitor nirogacestat, have had the most promising results in randomized placebo-controlled trials [2], although objective imaging responses rates can remain low [3]. Cryoablation has also emerged as a frontline option to control disease and improve symptoms, with one-year progression-free survival (PFS) of 86% [4]. Given the variety of treatment options, assessment of treatment response and surveillance for disease recrudescence make imaging a pillar for guiding management.

Commonly, size criteria based on Response Evaluation Criteria in Solid Tumors (RECIST) 1.1 are applied in clinical trials, but it is increasingly clear that size does not necessarily correlate with tumor biological response or patient reported outcomes [5]. In addition, the inherent variability between reviewers, and differences in imaging techniques and systems, can obfuscate slight changes in DTs. Newer efforts to determine treatment response include multiparametric imaging techniques, higher order analyses such as radiomics, and more complex analysis techniques involving artificial intelligence (AI). A summary of recent developments is provided, along with helpful hints to more accurately assess DT tumor burden and therapeutic response, with the goal of ameliorating the “desmoid dilemma.”

## 2. Imaging of Desmoid Tumors

MRI remains the mainstay for imaging of DTs [6], although the use of CT [7], PET [7], and ultrasound [8,9] has been reported (Table 1). In general, thin slices (5 mm slice thickness or thinner) with administration of intravenous contrast are advised, preferably with venous phase timing. On MR, fat suppression is recommended on contrast-enhanced sequences; additional T1 sequences may be performed for the assessment of fat planes around neurovasculature in conjunction with fluid-sensitive (fat-suppressed proton density (PD) or T2 imaging) sequences. Advanced MRI sequences, such as diffusion-weighted imaging (DWI), T2 mapping, dynamic contrast-enhanced imaging, elastography, etc., can provide additional information on tumor status as described in later sections, but their performance is institution dependent.

DTs often display heterogenous appearances on MRI ultimately owing to variable cellularity, collagen content, and myxoid stroma (tissue with complex mucopolysaccharide sugars) [10]. Prominent areas of collagen will appear as nonenhancing T1/T2 hypointense curvilinear or swirling bands, a characteristic but not pathognomonic imaging feature. Conversely, areas of cellularity and myxoid matrix will show intermediate to low T1 and intermediate to high T2 signal intensity, with variably avid contrast enhancement [6]. This appearance can differ between primary and recurrent DTs; primary tumors demonstrate heterogenous enhancement with a fascicular morphology, and recurrent tumors are more varied with either homogenous or heterogenous enhancement and a predominately fascicular morphology, although ovoid, nodular, streaky, and polycyclic configurations are possible [11].
curroncol-32-00288-t001_Table 1Table 1Comparison of imaging modalities used for desmoid tumors.ModalityAppearanceStrengthsLimitationsMRIWell-circumscribed, heterogenous with nonenhancing T1/T2 hypointense curvilinear areas related to collagen, and T1 intermediate to low/T2 intermediate to high areas related to cellularity or the myxoid matrix. Additional secondary signs (“flame”, “staghorn”, “fascial tail”, etc.) are described in text.-Suited for evaluation for treatment monitoring and planning (e.g., visualization of local extension and invasion of adjacent structures)-Opportunity for multiparametric approach including specialized sequences for functional imaging-Higher contrast than CT-Lack of radiation-Lower spatial resolution than CT-Higher costs with longer imaging timesCTWell-circumscribed, relatively homogenous soft tissue attenuation (generally hypodense to muscle) with variable enhancement.-Suited for evaluation for treatment monitoring and planning, particularly for deeper DTs-Higher spatial resolution than MRI-Often used in tandem during biopsies-Lower soft tissue contrast compared to MRI-Radiation risks, particularly in serial imagingUSSmooth, well-defined margins with homogenous hypoechoic echogenicity. Echogenicity and vascularity can vary depending on internal composition. Borders can also be ill-defined or irregular.-Safe in pregnancy-Fast, lower-cost compared to CT or MRI for initial imaging of superficial DTs-Contrast enhanced ultrasound is being explored for treatment monitoring [12]-Limited depth-Limited evaluation for treatment planning (e.g., visualization of local extension and invasion of adjacent structures)-Operator dependentPETMild to moderate uptake with median SUV of 3.1 (range 2.0–7.3) [7].-Lower SUV can suggest diagnosis of a benign process such as DT, but a specific discriminatory SUV cutoff has not been defined-Decrease in SUV on sequential scans may indicate treatment response in known DTs-Limited evidence for use in literature-Higher cost of PET/CT than traditional CT-Nonspecific for DTs

Depending on the location of the DT, additional secondary signs may be present. In extra-abdominal DTs, extension to adjacent subcutaneous fat or muscle can proceed along multiple fibrous bands, producing a “staghorn” appearance (”staghorn sign”), or along a muscle fiber or aponeurosis, creating a linear extension termed the “fascial tail sign” analogous to the “dural tail sign” seen with meningiomas [13]. DT can be found purely within or spanning multiple levels from the subcutaneous, muscular, and fascial layers. Superficial and deep intermuscular tumors with prominent fascial infiltration have a higher likelihood of treatment failure and postoperative recurrence than those within muscle [14,15]. For intermuscular DTs, there may be a thin rim of surrounding fat with feathery/flamelike margins (“flame sign”). In contrast, intraabdominal DTs, which are commonly located in the mesentery, often display a whorled appearance with ill-defined margins, representing a spiculated pattern of extension into the adjacent mesenteric fat. The irregular shapes and infiltrative margins displayed by DTs contribute to the difficulty in reliably measuring them in both 3D and single plane techniques (Figure 1). While the correlation is not perfect, the percentage of tumor volume that is T2 hyperintense likely has prognostic importance in predicting tumors that are at risk for progression with observation only [16,17]. One retrospective study of 37 patients managed with observation showing that ≥90% volume of T2 hyperintensity is associated with 1-year PFS of only 55% (*n* = 20), compared with 94% in the <90% group (*n* = 17) [17].

While imaging may be helpful for treatment planning and monitoring, diagnosis requires pathological confirmation, and core needle biopsy remains the gold standard [18]. On histopathology, DTs appear as interlacing bundles of elongated spindle cells in a collagen matrix, with a varying composition of spindle cells and collagen. To differentiate DTs from other soft tissue tumors, immunohistochemistry further demonstrates staining of certain proteins, such as vimentin and smooth muscle actin, in contrast to other immunohistochemical markers, such as desmin, S-100, or cytokeratins [19]. Beta-catenin immunohistochemistry separates desmoid fibromatosis from entities in the differential diagnosis, a finding that can be exploited for diagnosis [20]. DTs also carry mutations in the Wnt/APC/β-catenin pathway, which controls embryonic and embryonal stem cell development, with sporadic DTs comprising the overwhelming majority (85–90%), demonstrating β-catenin mutations, and inherited forms (10–15%) demonstrating APC mutations as part of Gardner’s syndrome, a subtype of Familial Adenomatous Polyposis [21]. However, even with these potential findings, misdiagnoses of DTs are common and affect up to 30–40% due to their rarity and similar appearance to other myofibroblastic diseases [21], including low grade soft tissue neoplasms [9], and a second opinion by another pathologist is recommended.

## 3. Current State of Evaluating Treatment Response

DT clinical trials have most commonly employed size-based criteria through the RECIST 1.1 framework to determine an objective response [22], although other response criteria have been explored in the literature (Table 2). Accordingly, treatment response is categorized into either complete response, partial response, stable disease, or progressive disease when comparing the baseline and follow-up appearance of “target lesions,” defined as up to five lesions, with a maximum of two lesions per organ. Because desmoids do not metastasize, lymph nodes should not be included as target or non-target lesions. Measurements should generally be performed in the axial plane, unless better visualized on a different plane, with the same plane used in follow-up evaluations for consistency. Whereas a complete response represents the disappearance of all lesions, a partial response represents a ≥30% decrease in the sum of the longest diameters (SLD) of target lesions without new lesions or progression of non-target lesions; progressive disease represents a ≥20% increase in the SLD compared to nadir SLD during treatment, or the presence of new lesions or unequivocal progression of non-target lesions. Stable disease is a catch-all categorization of the remaining cases.

For DTs specifically, care should be exercised to include all tumor components, which include enhancing, non-enhancing hypointense and T2 hyperintense tissue. DTs commonly have an infiltrative, irregular morphology, which may involve lesions with thin connective tails. To facilitate measurements, it is recommended that lesions connected by tails less than 2 mm be considered discontinuous separate entities.

RECIST can be criticized as being complicated with arbitrary thresholds chosen for response versus progression, with both intra-reader and inter-reader variability affecting the validity [14]. Multiple variations of RECIST and other methods including the European Association for the Study of the Liver (EASL) criteria, Cheson criteria, Choi and modified Choi criteria have been developed for tumor response evaluation in other neoplasms and applied to DT monitoring [23]. Some authors have noted the inter-rater and intra-rater reliability for DT measurement to be so poor that attempts at more comprehensive measurement techniques, such as 2D area and 3D volumetric measurements, had to be abandoned [14]. However, size-based criteria are still commonly used. Advantages include compatibility with conventional MR sequences without need for 3D isotropic acquisitions, avoidance of signal ratio calculations, and limited influence from different scanners and imaging protocols. Volumetric analyses also minimize the impact of fascial tails on size measurements, with some even applying ellipsoid approximations for volume [24] (although new errors can be introduced from the approximation). It may be that greater thresholds are more appropriate for the determination of treatment response, with some estimates that progression can only be reliably identified with volume increases greater than 42.6% after accounting for variability from inter-reader measurements and differences in MRI in-plane resolution [24]. Indeed, Subhawong et al. demonstrated earlier detection of a partial response using volumetric decreases of greater than 50% compared to RECIST (mean time of 1.1 years versus 2 years), with earlier or contemporaneous detection in each of the 10 partial responders defined by RECIST [25].

Aside from the inherent difficulty in obtaining reliable DT measurements for RECIST, this and other conventional imaging response categorization techniques do not account for detectable physiological changes distinct from size, namely MRI signal changes [1,26]. Successfully treated and quiescent DTs have long been known to become fibrotic and collagenized in cellular areas, decreasing both the T2 signal and enhancement (Figure 2 and Figure 3) [6]. To account for this, Gounder et al. proposed quantifying treatment response as a function of the ratio of the mean intensity of DTs at the largest cross-sectional diameter with adjacent muscle and a more semi-quantitative metric where radiologists estimated the percentage, in 10% increments, of the tumor that demonstrated less signal intensity than muscle on T2 imaging [23]. Subhawong et al. proposed a variation where the ratio of the mean intensity of the entire volume of the DT on T2 imaging was compared to muscle (tumor to muscle ratio) [25]. Furthermore, there is no consensus threshold ratio for determining PD and PR; some authors have proposed T2 tumor to muscle ratios as low as one to guide decision-making [24], whereas others have opted for RECIST-aligned 20% increase/30% decrease for PD/PR [25,27].
curroncol-32-00288-t002_Table 2Table 2Comparison of commonly used imaging response criteria in clinical trials for DTs.CriteriaPartial ResponseProgressive DiseaseRECIST 1.1≥30% decrease in sum of longest diameters (SLD) of target lesions without new lesions or progression of non-target lesions≥20% increase and ≥5 mm absolute increase in SLD of target lesions or presence of new lesions or progression of non-target lesionsmodified RECIST≥30% decrease in SLD of contrast-enhancing areas of target lesions≥20% increase in SLD of contrast-enhancing areas of target lesionsWHO≥50% decrease in sum of the product of the diameters (SPD) of lesions≥25% increase in SPD of at least one of the lesions or presence of new lesionsChoi≥10% decrease in SLD of target lesions or ≥15% decrease in attenuation on CT, without new lesions or progression of nonmeasurable disease≥10% increase in SLD of target lesions or increase/<15% decrease in attenuation on CT or new lesions or progression of nonmeasurable diseasemodified Choi≥10% decrease in SLD of target lesions AND ≥15% decrease in attenuation on CT, without new lesions or progression of nonmeasurable disease≥10% increase in SLD of target lesions AND increase/<15% decrease in attenuation on CT or new lesions or progression of nonmeasurable diseaseMRI-based modified Choi [25,28]≥10% decrease in SLD of target lesions AND ≥30% decrease in volumetric tumor-to-muscle signal ratio from baseline, without new lesions or progression of nonmeasurable disease≥10% increase in SLD of target lesions AND >20% increase in volumetric tumor-to-muscle signal ratio from baseline or new lesions or progression of nonmeasurable diseaseVolumetric [25,28]>50% decrease in total volumetric size>25% increase in total volumetric sizeEASL≥50% decrease in total tumor load>25% increase in size of one or more lesions or presence of new lesionsCheson≥50% decrease in SPD from baseline≥50% increase in SPD from baseline or ≥50% increase in longest diameter of any lesion or presence of new lesionsFor all criteria, complete response is represented by disappearance of all lesions, and stable disease is represented by all cases not matching a complete response, partial response, or progressive disease.

In general, patients that were considered to have a treatment response through RECIST also demonstrated MRI signal changes, although a greater number of patients were considered to have treatment response with MRI signal changes, and the MRI signal changes often preceded the threshold size changes required for RECIST [23,25,29]. However, typical signal changes vary across treatments; for example, systemic therapy often induces collagenization, whereas cryoablation creates tumor necrosis, resulting in different MR appearances (T2 hypointensity versus hyperintensity, respectively), as shown in Figure 4. Tumor enhancement, however, generally decreases in the setting of treatment response, irrespective of treatment modality, and may represent a more attractive imaging endpoint, although there is a lack of consensus around appropriate PR and PD thresholds. Whether one- or two-dimensional measurements or signal ratios of enhancing tumor components are used depends on the response criteria selected, though recently they were all shown to perform poorly when correlated with PFS [30].

## 4. Experimental Approaches

With the rise of radiomics, higher order statistics have been applied to DT treatment response determination. Increased tumor heterogeneity assessed through changes in texture from baseline (delta radiomics) on T2 imaging correlates with a poor response [25,31] extending to worse patient outcomes, including decreased progression-free survival (PFS) [31]. Crombe et al. also found baseline and delta radiomic features derived from fat-suppressed contrast-enhanced T1 imaging were associated with PFS in chemotherapy-treated desmoid tumors, while some size-based response criteria did not correspond to PFS outcomes in this population [30,32]. Consequently, radiomics offers the potential for earlier detection of a treatment response and correlation with patient outcomes, although the results are preliminary and feature stability will require validation in subsequent studies.

Additional routinely performed and commercially available MRI techniques may also provide new methods to characterize DTs. DWI has been demonstrated to show higher apparent diffusion coefficient (ADC) values in DTs versus malignant soft tissue tumors. The potential applicability of DWI for assessing DT response may be confounded by necrosis following some therapies, such as cryoablation, resulting in decreased ADC signal, which can mimic a viable or recurrent tumor [33]. Another area of interest is T2 mapping (Figure 5), which quantifies absolute tissue T2 relaxation times and surmounts some of the inherent limitations of using tumor to muscle T2 signal ratios, with high inter-reader reliability [34]. T2 mapping has been shown to correlate with non-perfused volume to assess the reduction in viable tumor volume following therapeutic ablation by Morochnik et al. [35]. Finally, a pilot study on 10 tumors demonstrated earlier detection of progression by 4.5 months using volume and signal changes on contrast-enhanced susceptibility-weighted imaging (CE-SWI) compared to T2-based RECIST 1.1 criteria, with greater sensitivity toward collagenized areas in tumors compared to T2 imaging [28]. CE-SWI is a potentially beneficial single sequence that could provide a combination of the tissue change characteristics traditionally derived from separate T2 and postcontrast sequences. Additionally, the more pronounced soft tissue contrast with the CE-SWI technique produces greater changes in first-order radiomic features to discriminate between responding and non-responding tumors when compared to radiomics derived from standard T2-weighted imaging. Although these sequences may be available on clinical MRI systems, there is an increased cost for specialized protocols, which can be on the order of hundreds of dollars for even a single added sequence [36,37]. As advanced imaging techniques continue to be explored for monitoring DT treatment response, the frequency of follow-up may need to be adjusted according to patient risk, especially in clinical settings where cost concerns are more prominent [38].

## 5. Persistent Challenges to Evaluation of Treatment Response

As more novel methods of DT response evaluation emerge, a validated “gold standard” for response assessment remains elusive. The RECIST criteria are familiar and have been the default choice to determine objective response rates in the majority of prospective DT clinical trials, despite the known shortcomings. While recent studies frequently include alternatives such as the volumetric and modified Choi, and others as previously discussed, all these similar and somewhat overlapping methods are grouped into a “conventional” imaging response criteria category [9]. This adds both complexity and potentially less certainty when evaluating new metrics, as specific imaging features may correspond to different response categories depending on the exact conventional imaging response criteria selected.

Additional DT response challenges are universal to any methodology, including how to assess multiple lesions by either aggregating the behavior of all lesions or selectively omitting measurements of some lesions, and DT-specific challenges such as accounting for intra-tumor heterogeneity in biologic aggressiveness. This dilemma adds complexity and difficulty in conforming to RECIST 1.1 [39]. Additionally, differences in the MRI protocol can cause variation in reliability between and within patients over time, limiting generalization across centers. Approaches using skeletal muscle MR signal as a normalization factor have been popularized, but muscle signal can demonstrate heterogeneity within a designated region of interest and across different muscle groups due to natural variation, sarcopenia (loss of muscle mass and strength with age), denervation changes, and inflammation, among other etiologies. Additional nonuniformity can result from imaging artifacts, including inhomogeneous fat suppression and dielectric effects [34]. Finally, tumor response is traditionally focused on changes intrinsic to the tumor, without regard to implications of tumor invasion into (or regression away from) adjacent critical structures or organs. Such changes can become important for prognosis, particularly with DTs where nerve or vessel compromise may necessitate surgical intervention while simultaneously increasing the risk of recurrence if critical structure involvement precludes a wide resection margin. Accounting for these challenges is important to consider during assessment of DT response and in the pursuit of a more comprehensive response evaluation framework.

## 6. Conclusions

Desmoid tumors present many challenges to treatment, reliable imaging characterization, and biologic behavior prediction. DT response to therapy assessment on imaging is complicated by distinct biological and behavioral characteristics, namely its mixed appearance caused by the variable proportions of cellular and collagenized tissue components, and the spectrum of morphological changes in treatment response beyond a simple change in size. The incorporation of MRI signal changes has thus become integral to evolving research pursuing novel metrics to supplement or supplant traditional size-based response metrics. Several proposed methods, most of which emphasize T2 and enhancement characteristics, have shown good correlation to the traditional RECIST approach, while offering earlier response detection by considering the functional tissue changes as the treated tumor becomes less cellular and more collagenized. Experimental approaches through new analyses, such as radiomics, and functional and parametric imaging methods such as diffusion-weighted MRI, dynamic contrast enhanced MRI, and T2 mapping, may continue to increase the sensitivity of detection of the physiological changes occurring in treated DTs.

The infiltrative nature and irregular shape of many DTs creates a significant challenge for reliability in size measurements, which are foundational to conventional imaging response criteria. Other barriers to reproducibility include differences in MRI protocols and technologic factors, which may strongly impact emerging methods that rely on advanced quantitative feature analysis. Future studies of novel response criteria would ideally be correlated with DT-specific patient reported outcome measures, such as the validated GOunder/Desmoid tumor research foundation and DEsmoid Symptom/impact Scale (GODDESS). Continued refinement of MRI signal change approaches, along with possible future consensus methods, may reduce the limitations and provide clarity to help align imaging surveillance with true treatment efficacy and ultimately improve outcomes for patients with desmoid tumors.

## Figures and Tables

**Figure 1 curroncol-32-00288-f001:**
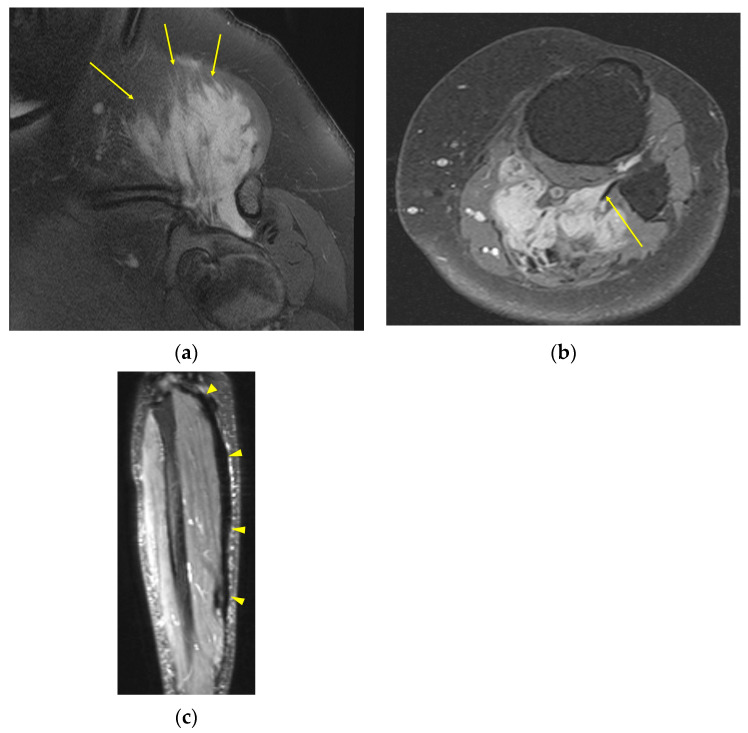
Morphologic features of DTs that can be challenging to accurately measure on T2-weighted MR, including (**a**) infiltrative margins in a DT above the shoulder joint (arrows), (**b**) “fascial tail” extension along the popliteus muscle in this patient with DT of the left calf musculature (arrows), and (**c**) elongated, plaque-like morphology extending along the superficial gastrocnemius (arrowheads).

**Figure 2 curroncol-32-00288-f002:**
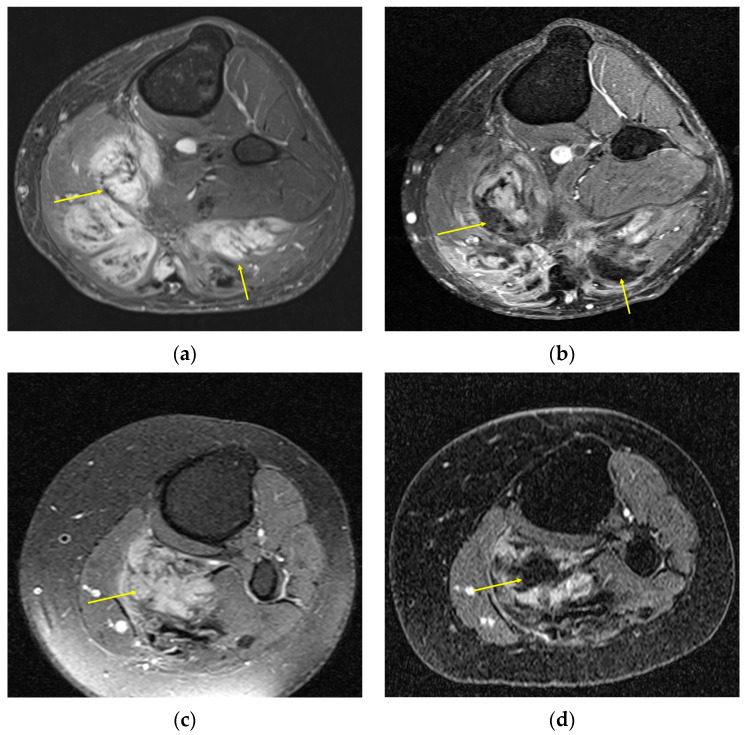
Top row, a 23-year-old man with a left popliteal mass DT, pretreatment (**a**) and posttreatment (**b**) axial MRI T1 postcontrast with fat saturation images showing multifocal interval decrease in contrast enhancement (arrows), despite grossly stable size of tumor. Bottom row, a 58-year-old woman with a similar location DT, pretreatment (**c**) and posttreatment (**d**) axial fluid sensitive fat-suppressed MRI images showing interval loss of central T2 hyperintense signal (arrow). Both patients were classified by RECIST as stable disease due to a lack of size change.

**Figure 3 curroncol-32-00288-f003:**
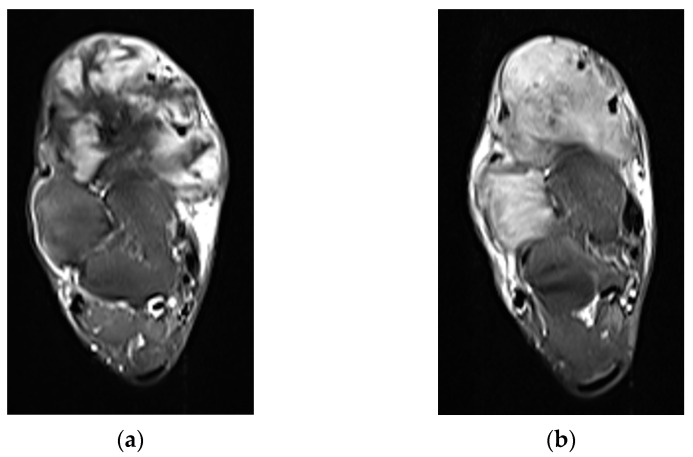
An 18-year-old woman with a right anterior foot DT, 6 months posttreatment with nirogacestat, a γ-secretase inhibitor, (**a**,**c**) and pretreatment (**b**,**d**) axial MRI T2 fat-suppressed images in the top row and sagittal MRI T2 fat-suppressed images in the bottom row, showing an interval decrease in T2 signal, suggestive of a treatment response.

**Figure 4 curroncol-32-00288-f004:**
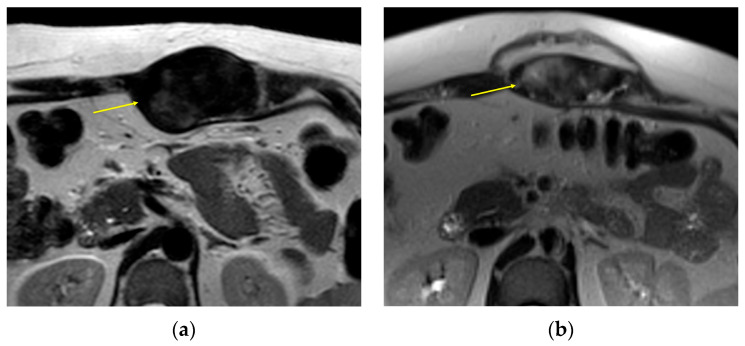
A 36-year-old woman with an abdominal wall desmoid tumor. Pretreatment T2 images (**a**) were predominantly hypointense suggesting low cellularity. The patient underwent cryoablation therapy shown in axial CT (**c**) with ablation probe (arrow) and subcutaneous gas (arrowheads). Posttreatment T2 (**b**) shows heterogeneous increased internal T2 signal (arrow), which could be mistaken for increased cellularity. However, comparison of the preablation (**d**) and postablation (**e**) postcontrast T1 images show marked reduction in enhancement with increased internal fluid levels (arrow), compatible with post-ablation necrosis.

**Figure 5 curroncol-32-00288-f005:**
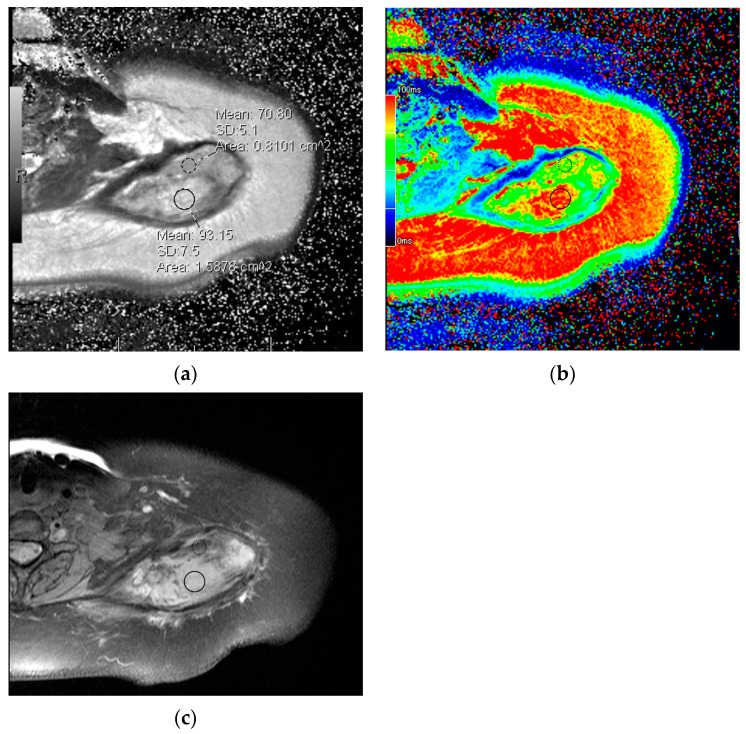
A 32-year-old woman with a DT centered in the left supraspinatus muscle. Axial images with T2 mapping (**a**) show some areas of high T2 relaxation time, corresponding to increased cellularity with warmer values on the color map (solid black circle, **b**) and hyperintense signal on a typical fat-suppressed MR image (solid black circle, **b**). An adjacent area of low-intermediate T2 relaxation time is shown for comparison (dashed circle in **a**,**c**).

## Data Availability

Not applicable.

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
