# Peer review of "The Desmoid Dilemma: Challenges and Opportunities in Assessing Tumor Burden and Therapeutic Response"

_curroncol, 2025, doi:10.3390/curroncol32050288_

Round 1
Reviewer 1 Report
Comments and Suggestions for Authors
The manuscript addresses a clinically significant and under-discussed issue in oncology—the complexity of assessing desmoid tumor response to therapy using conventional imaging criteria. The integration of advanced imaging modalities and AI approaches is timely and offers valuable insights for improving clinical decision-making.
However, I have some suggestions for improvement:
- Provide a summarized comparison table of the imaging modalities and their corresponding strengths/limitations in DT assessment.
- Clarify terminology in some technical descriptions for a broader audience, particularly radiologists who may not specialize in soft tissue tumors.
- Consider including a short discussion on cost/accessibility of advanced imaging techniques in routine clinical settings.
- Please add the following references to your imaging part: DOI: 10.1055/a-1022-4546
Author Response
The manuscript addresses a clinically significant and under-discussed issue in oncology—the complexity of assessing desmoid tumor response to therapy using conventional imaging criteria. The integration of advanced imaging modalities and AI approaches is timely and offers valuable insights for improving clinical decision-making.
However, I have some suggestions for improvement:
Provide a summarized comparison table of the imaging modalities and their corresponding strengths/limitations in DT assessment.
Table 1 has been added to summarize imaging modalities and their strengths and limitations in DT assessment.
Clarify terminology in some technical descriptions for a broader audience, particularly radiologists who may not specialize in soft tissue tumors.
Technical descriptions and jargon throughout the manuscript have been clarified where possible with short explanations where needed.
Consider including a short discussion on cost/accessibility of advanced imaging techniques in routine clinical settings.
A short discussion on the cost and accessibility of advanced imaging techniques in routine clinical settings has been added to the end of the “Experimental Approaches” section. Relative cost comparison has also been added to the strengths/limitations of imaging modalities in Table 1.
Please add the following references to your imaging part: DOI: 10.1055/a-1022-4546
The reference has been added.
Reviewer 2 Report
Comments and Suggestions for Authors
This is a review article with clinical significance. It clearly conveys the dilemma of image evaluation of desmoid tumors, especially in determining treatment efficacy. The literature selected and the descriptions are appropriate, with no significant problems.
I think a table summarizing the endpoints used in various clinical trials would be more useful for readers. Please summarize what evaluation methods are used for image evaluation and what evaluation methods other than RECIST are being explored.
Author Response
This is a review article with clinical significance. It clearly conveys the dilemma of image evaluation of desmoid tumors, especially in determining treatment efficacy. The literature selected and the descriptions are appropriate, with no significant problems.
I think a table summarizing the endpoints used in various clinical trials would be more useful for readers. Please summarize what evaluation methods are used for image evaluation and what evaluation methods other than RECIST are being explored.
As recommended, Table 2 has been added to summarize current and exploratory image evaluation methods including and other than RECIST1.1.
Reviewer 3 Report
Comments and Suggestions for Authors
The findings of this study will be of significant interest to health professionals. However, core biopsy is considered the gold standard for diagnosing soft tissue tumors, including desmoid fibromatosis. Without confirmation of the diagnosis by an expert in bone and soft tissue pathology, there may be uncertainty in the accuracy of the diagnosis, which could influence treatment decisions. Please provide additional information about the diagnostic approach. Histology examination/diagnosis?
Author Response
The findings of this study will be of significant interest to health professionals. However, core biopsy is considered the gold standard for diagnosing soft tissue tumors, including desmoid fibromatosis. Without confirmation of the diagnosis by an expert in bone and soft tissue pathology, there may be uncertainty in the accuracy of the diagnosis, which could influence treatment decisions. Please provide additional information about the diagnostic approach. Histology examination/diagnosis?
It is now made clear in a new paragraph added to the "Imaging of Desmoid Tumors" section that core biopsy is the gold standard for diagnosis. The histopathologic, immunohistochemical, and genetic pattern of desmoid tumors is added along with a brief discussion on the difficulty of diagnosis.
Round 2
Reviewer 3 Report
Comments and Suggestions for Authors
The authors have revised the manuscript; however, a minor revision is still required regarding the immunohistochemistry. Please see below.
Page 4 line 119-122: "To differentiate from other soft tissue tumors, immunohistochemistry further demonstrates staining of certain proteins, such as vimentin and smooth muscle actin, in contrast to other immunohistochemical markers, such as desmin, S-100, or cytokeratins [19]."
Please add after [19] "Beta-catenin immunohistochemistry separates desmoid fibromatosis from entities in the differential diagnosis, a finding that can be exploited for diagnosis (PMID: 15832090)."
Author Response
Comments 1:
The authors have revised the manuscript; however, a minor revision is still required regarding the immunohistochemistry. Please see below.
Page 4 line 119-122: "To differentiate from other soft tissue tumors, immunohistochemistry further demonstrates staining of certain proteins, such as vimentin and smooth muscle actin, in contrast to other immunohistochemical markers, such as desmin, S-100, or cytokeratins [19]."
Please add after [19] "Beta-catenin immunohistochemistry separates desmoid fibromatosis from entities in the differential diagnosis, a finding that can be exploited for diagnosis (PMID: 15832090)."
We thank the reviewer for their comment. As recommended, the sentence and corresponding reference has been added to the manuscript.